# Influence of Curly Leaf Trait on Cottonseed Micro-Nutrient Status in Cotton (*Gossypium hirsutum* L.) Lines

**DOI:** 10.3390/plants10081701

**Published:** 2021-08-18

**Authors:** Nacer Bellaloui, Rickie B. Turley, Salliana R. Stetina

**Affiliations:** Crop Genetics Research Unit, USDA, Agricultural Research Service, 141 Experiment Station Road, Stoneville, MS 38776, USA; rick.turley@usda.gov (R.B.T.); sally.stetina@usda.gov (S.R.S.)

**Keywords:** cottonseed, cottonseed nutrition, minerals, nutrients, cottonseed feed, leaf morphology

## Abstract

Cottonseed is a source of nutrients, including protein, oil, and macro- and micro-nutrients. Micro-nutrients such as boron (B), copper (Cu), iron (Fe), manganese (Mn), and zinc (Zn) are essential for plant and human health. Deficiencies of these micro-nutrients in soil lead to poor crop production and poor seed quality. Micro-nutrient deficiencies in the human diet lead to malnutrition and serious health issues. Therefore, identifying new cotton lines containing high nutritional qualities such as micro-nutrients, and understanding plant traits influencing micro-nutrients are essential. The objective of this research was to investigate the effects of leaf shape (curly leaf: CRL) on cottonseed B, Cu, Fe, Mn, Ni (nickel), and Zn in two near-isogenic cotton lines differing in leaf shape (DP 5690 wild-type with normal leaves and DP 5690 CRL). We also used Uzbek CRL, the source of the curly leaf trait, for comparison. A field experiment was conducted in 2014 and 2015 in Stoneville, MS, USA. The experiment was a randomized complete block design with three replicates. The results showed that, in 2014, both DP 5690 wild-type and Uzbek CRL had higher seed B, Cu, Fe, and Ni than in DP 5690 CRL. The accumulation of Mn and Zn in seeds of DP 5690 CRL was higher than in DP 5690 wild-type and Uzbek CRL. However, in 2015, the concentrations of B, Cu, Fe, and Ni, including Mn and Zn, were higher in both DP 5690 wild-type and Uzbek CRL than in DP 5690 CRL. Positive and negative correlations existed in 2014; however, only positive correlations existed between all nutrients in 2015. This research demonstrated that leaf shape can alter cottonseed micro-nutrients status. As Uzbek CRL behaved similar to wild-type, both leaf shape and other factors contributed to the alteration in seed micronutrients, affecting seed nutritional qualities. Therefore, leaf-shape partially contributed to the changes in micro-nutrients in cottonseed. The negative and positive correlations in 2014, and only positive correlations in 2015, were likely due to the heat difference between 2014 and 2015 as 2015 was warmer than 2014. Significant levels of seed micro-nutrients were shown between these lines, providing opportunities for breeders to select for high seed micro-nutrients in cotton. Additionally, the current research provides researchers with physiological information on the impact of leaf shape on seed nutritional quality. The leaf shape trait can also be used as a tool to study leaf development, physiological, biochemical, and morphological processes.

## 1. Introduction

Micro-nutrients are essential for plant and human health. Deficiencies of micro-nutrients in soil result in yield loss and poor seed nutritional quality [1,2]. Micronutrient deficiencies of Fe and Zn resulted in serious health issues in pregnant women and children [3]; Fe deficiency resulted in anemia of the preschool-aged children in Africa and southeast Asia [4]; and Zn deficiency was reported to be a public health risk, especially for children under the age of five [5].

Therefore, maintaining optimum levels of micro-nutrients in seeds, or identifying new genotypes with higher micro-nutrient content in crop seed, is essential for human and livestock nutrition. Although micro-nutrients are only needed in small amounts by crops, their roles in growth, development, function, and seed health are essential. Previous research indicated the significance and involvement of micro-nutrients B, Zn, Fe, Cu, Mn, and Ni in the function of enzymes and hormone synthesis; photosynthesis and chlorophyll function (Mg), lipids (P), carbohydrate translocation (B), and nitrogen metabolism (Fe) [6,7]. The accumulation of minerals in seed involves physiological, genetic, and environmental factors controlling nutrient uptake, translocation, redistribution, and accumulation [8,9], and most of the genetic bases of these process are not well understood [10,11].

Cotton is a major crop in the world [12] and cottonseed is considered a major source of fiber, oil, proteins, and macro- and micro-nutrients for human food products and animal feed [12,13,14,15,16,17]. Cottonseed nutritional qualities have been relatively neglected, narrowing the genetic variation of cotton for seed quality improvement through breeding. Therefore, identifying other cotton genotypes that influence the accumulation of micro-nutrients in seeds that determine cottonseed nutritional qualities is crucial [18,19,20,21]. As the leaves are the main site for photosynthesis, nutrient metabolism, and metabolite synthesis, the current research will investigate the effects of leaf shape (curly leaf) trait on cottonseed micro-nutrients accumulation. Most curly leaf studies conducted to date were related to cell division, leaf development, and lint yield, but information on cottonseed nutrients, including micro-nutrients, is almost non-existent.

Understanding the molecular processes controlling leaf shape phenotype may help advance our efforts in developing cotton cultivars with ideal leaf shape, enhancing sustainability, profitability, and production [22]. Studies were conducted on major leaf shapes in Upland cotton and its multiple allelic of a single incomplete dominant genetic locus *L-D1* on chromosome 15-D1 (Chr15), and major leaf shape genes mapping in cotton and gene expression changes [22]; on cultivars of normal leaves or broad leaves (NL), sub-okra/Sea-Island (subOL), Okra (OL), Superokra (superOL), leaf shapes, cotton disease, insect resistance, yield production, and fiber quality [23,24,25,26,27,28,29].

Regulation of curly leaf gene, a histone methyltransferase of Polycomb Repressive Complex 2 (PRC2) for trimethylation of histone H3 Lys 27 (H3K27me3), was studied by others [30]; the *Arabidopsis* FERTILIZATION-INDEPENDENT ENDOSPERM (FIE) polycomb group protein regulated the development of endosperm and embryo, and repressed the flowering during embryo and seedling development, as studied by others [31]. Studies on a gene encoding an E3 ligase (UPWARD CURLY LEAF1 (UCL1), which degrades the CURLY LEAF (CRL) polycomb protein, a PEG, were conducted by others [32].

The objective of the current research was to investigate the influence of leaf shape (curly leaf trait) on cottonseed micro-nutrients content using two isogenic cotton lines differing in leaf shape (curly leaf) (DP 5690 wild-type with normal leaf and DP 5690 CRL; Uzbek CRL was used as parent and source of CRL). We hypothesized that, as the leaf is a main site of nutrient accumulation and related to nutrient uptake, assimilation, and photosynthesis process, cottonseed micro-nutrients content will differ between these lines with different leaf shapes.

## 2. Results and Discussion

ANOVA showed that year, line, and their interactions were significant for micro-nutrients, indicating that these factors are the main source of variability for these micro-nutrients (Table 1). ANOVA showed that year and line were significant for all micro-nutrients, except the line for Zn. Additionally, year x line interactions were significant for Mn and Zn. The different response of macro-nutrients in lines in each year could be due to temperature differences between years. For example, in 2014, the maximum temperatures were 31.39, 31.15, 32.49, and 31.28 °C, respectively, in June, July, August, and September; and in 2015 the maximum temperatures were 32.63, 34.10, 33.35, and 33.0 °C, respectively, in June, July, August, and September (Figure 1). It is clear that 2015 was warmer than 2014, and the higher temperature during these months (this period coincides with the critical reproduction stages from flowering, trough boll set, to seed-fill) can alter seed nutrients, as high temperature was reported to affect the uptake, the mobility of nutrients from leaves (source) to seed (sink), and the accumulation of nutrients in seeds [33,34,35,36]. The experiment was irrigated in both years to supplement rainfall and mitigate drought stress, so differences in soil moisture are not a likely source of variability.

In 2014, mean values showed that Fe, B, Cu, and Ni were higher in both DP 5690 wild-type and Uzbek CRL than in DP 5690 CRL. The concentration of Mn and Zn were higher in DP 5690 CRL than Uzbek CRL and wild-type DP 5690 (Table 2). In 2015, the concentration of Fe, B, Cu, and Ni were higher in both DP 5690 wild-type and Uzbek CRL than in DP 5690 CRL. The concentration of Mn and Zn were also higher in DP 5690 wilt-type and Uzbek CRL than DP 5690 CRL, opposing that which occurred in 2014 (Table 3). The accumulation of nutrients was higher in 2014 than in 2015.

In 2014, correlation between nutrients across lines showed that positive correlation existed between B and Fe and between B and Ni; a negative correlation existed between B and Zn. A positive correlation existed between Cu and Fe; a negative correlation existed between Cu and Mn and between Cu and Zn. Iron was positively correlated with Ni and negatively correlated with Zn (Table 4). In 2015, a positive correlation was observed between all pairs of studied micro-nutrients (Table 5). Micro-nutrient concentrations varied significantly among the lines, although the frequency distributions varied from one micro-nutrient to the next and from one year to the next. (Figure 2, Figure 3, Figure 4 and Figure 5). DP 5690 wild-type and Uzbek CRL had larger seeds than DP 5690 CRL, and the 100-seed weight was 12.04 g in Uzbek CRL line, 10.43 g in DP 5690 wild-type, and 9.48 g in DP 5690 CRL. Similarly, Uzbek CRL line had the largest boll, followed by the DP 5690 wild-type, and then DP 5690 CRL with the smallest size (Figure 6). Visual growth rate and biomass observation in the field of the three lines (Figure 7) showed that DP 5690 CRL had the lowest growth rate and DP 5690 wild-type had the highest.

The significant effects of year and lines indicate that these factors are the main source of variability for these nutrients. These findings are in an agreement with previous studies on cottonseed and other species such as soybean, corn, and sesame [37,38,39]. The significant effects of the interaction of year with line indicated that the environmental conditions such as heat and drought, were different in each year, influencing the ranking of these nutrients in each year.

The higher concentrations of B, Cu, Fe, and Ni in DP 5690 wild-type compared with DP 5690 CRL could be due to the shape of the leaf (the trait), as DP 5690 wild-type and DP 5690 CRL are near isogenic. As Uzbek CRL behaved similar to DP 5690 wild-type, the differences in nutrients between DP 5690 wild-type and DP 5690 CRL could be partially due to the trait. The higher concentrations of Mn and Ni in DP 5690 CRL than DP 5690 wild-type and Uzbek CRL in 2014, and not in 2015, could be due, mainly, to higher heat in 2015.

Previous research showed that morphological, physiological, molecular, and genetic traits, including leaf shape, result in differences in cell division and elongation, leaf development, and cottonseed and fiber production and quality [22]. For example, cultivars with different leaf shapes of normal leaves or broad leaves (NL), sub-okra/Sea-Island (subOL), Okra (OL), and Superokra (superOL) resulted in differences in cotton disease infection, insect resistance, yield production, and fiber quality [22,23,24,25,26,27,28,29]. It was explained that the lower boll disease infection was due to microclimatic differences that led to canopy closure/openess, resulting for a greater air circulation and light penetration [22]. The morphology changes were due to the differential expression of alleles of the same genes in a parent-of-origin-specific manner. Others [32] studied the gene encoding an E3 ligase (UPWARD CURLY LEAF1 (UCL1), which degrades the CURLY LEAF (CRL) polycomb protein, a PEG, and found that the silencing of the maternal UCL1 allele was regulated by both cytosine DNA METHYLTRANSFERASE 1 (MET1) and the DNA glycosylase DEMETER (DME). Others [30] showed that *Arabidopsis* expressing *clf-28* produced bigger size and higher weight seeds with higher oil content, larger oil bodies, and altered long-chain fatty acid composition compared to wild-type, showing that curly leaf genes are involved in gene expression, and contribute to regulate specific physiological and molecular functions during embryo development and nutrient accumulation. We cannot suggest here that *Arabidopsis clf-28* genes behave as those of cotton, but we wanted to make this observation which may help the reader to learn.

Our research showed that B, Fe, Cu, and Ni were higher in both DP 5690 wild-type and Uzbek CRL than the DP 5690 CRL, indicating that the higher accumulation of these nutrients may reflect the bigger ball size and higher weight seed in both DP 5690 wild-type and Russian CRL. Although the mechanisms of how these nutrients involved in the curly leaf trait is still not well understood at the physiological, genetic, and molecular levels, the contribution of these nutrients to leaf photosynthesis, leaf growth and development, and seed quality was previously reported.

For example, B showed: (1) its involvement in RNA through uracil; cell division and differentiation; seed maturation; respiration; pollen germination, growth, and stability; and seed quality [7,18,19,40,41,42,43]; (2) involvement of Cu in chloroplast protein plastocyanin; electron transport system linking photosystem I and II; participation in protein and carbohydrate metabolism; involvement in enzymes such as oxidases for cytochrome, ascorbic, and polyphenol; and involvement in the desaturation and hydroxylation of fatty acids (7,40); (3) involvement of Mn in the oxidation reduction processes in the photosynthetic electron transport system, essential for photosystem II for photolysis; acting as a bridge for ATP and enzymes for phosphokinase and phosphotransferases; and activating IAA oxidases (7,40); (4) the involvement of Ni in disease resistance; nickel enzymes, proteins urease, and glyoxylase; and protein metabolism [7,40,44]. Previous research showed that micro-nutrients responded to phenotype traits, and micro-nutrients’ levels in plant tissues resulted in yield and seed quality differences [7,40]. For example, cotton NILs differing in seed fuzz [20,21] or leaf color [44] resulted in higher seed oil in fuzzless genotypes, but protein Ca and C were higher in fuzzy genotypes. Nutrients N, S, B, Fe, and Zn were higher in most of the fuzzy genotypes [20,21]. Cotton NILs differing in leaf color resulted in higher content of oil in green leaf lines than yellow leaf lines [44]. A similar trend was found for seed C, N, P, B, Cu, and Fe contents, where green lines had a higher content of these nutrients than yellow lines [44]. It can be concluded that morphological traits such as leaf shape can influence the regulation of nutrient uptake, translocation, and assimilation, leading to differences in micro-nutrient levels in cottonseeds between wild-type and CRL lines. The involvement of these micro-nutrients, especially in the carbon and nitrogen metabolism, photosynthesis process, enzymes activation, and oil production, such as B, Cu, Ni, and Zn, agree with the findings of others on cottonseed nutrients [20,21,22,30,32,44].

The positive and negative correlations between micro-nutrients could be due to growth conditions, genotype, and nutrient supply [7,18,19,40,41,42,43]. It was also reported that nutrient uptake, translocation, redistribution, and accumulation processes controlling the accumulation of nutrients in seeds [9,45], and their genetic basis are still not well understood [10]. The inconsistent correlations trend of some nutrients across years could be due to temperature differences in each year. Previous research showed that the correlations between seed nutrients in cotton and other crops changes from positive, to negative, to no-change, depending on the year, and was mainly attributed to gene x environment interactions [11,20,21,33,34,35,36,37,41,42,43]. The inter-relationships between nutrients and environment and genetics, and how they impact seed nutritional qualities, are still not well understood. Further research in this area is needed, as the relationship (negative, positive, or no-change) between nutrients is important as it impacts seed production and seed nutritional qualities. The phenomenon of ionic antagonism and synergism could be observed between the relationship between nutrients, for example, the synergetic relationship between all nutrients in 2014; however, only a few nutrients have antagonistic relationships in 2015. For example, the negative relationship between Mn and Cu; Zn and Cu; Mn and Fe; Zn and Fe; Ni and Mn; and Zn and Ni; all these antagonistic relationships were observed in 2015, and could be due to the environmental effects of heat as the year 2015 was warmer than 2014.

As the Uzbek line has curly leaves and behaved similar to the wild type, the differences in nutrients is not completely due to the leaf shape; rather, it could be due partially to leaf shape, and may be to the size of bolls, leaves, and plants, as shown in Figure 6 and Figure 7. The leaf size could be a source of differences in nutrient accumulation, as leaves of both DP 5690 wild-type and Uzbek CRL are bigger than the leaves of DP 5690 CRL, as shown in Figure 7. The combination of leaf shape and size could result in higher rates of photosynthesis, leading to bigger biomass, bigger boll size (Figure 6), and higher nutrient uptake and accumulation in cottonseeds.

## 3. Materials and Methods

### 3.1. Genetics and Growth Conditions of Cotton Lines

Cultivar DP 5690 (Bayer Corporation, Whippany, NJ, USA; PVPC 009100118) was developed into DP 5690 F_6_ by single seed descent for six generations. After three years of single seed descent, DP 5690 F_6_ (wild-type) with a theoretical purity of 98.44% was obtained. Then, the DP 5690 wild-type was crossed with the Uzbek CRL. After the resulted F_1_ progeny were selfed in the greenhouse once, DP 5690 wild-type was backcrossed to F_2_ plants with curly leaves. This process was repeated through six generations from BC_1_F_1_ through BC_5_F_2_, obtaining the normal and curly leaf phenotypes in a near-isogenic DP 5690 background.

A field experiment was conducted in Stoneville, MS, in 2014 and 2015 to study the influence of leaf shape trait (in our case curly leaf, CRL) on cottonseed micro-nutrients iron (Fe), boron (B), copper (Cu), zinc (Zn), and nickel (Ni) in a DP 5690 cotton (*Gossypium hirsutum* L.) background. Two near-isogenic lines (DP 5690 curly leaf and DP 5690 wild-type) and the Uzbek curly leaf parent were used. The experiment took place in single-row field plots spaced 1.02 m apart and 8.53 m in length, planted in a Bosket very fine sandy loam soil (fine loamy, mixed, active, thermic Mollic Hapludalfs) [46] on 5 May 2014 and 30 April 2015. Each line was replicated three times. To manage seedling diseases, pentachloronitrobenzene (Terraclor Super X 18.8 G, Chemtura USA Corporation, Middlebury, CT, USA) was applied in furrow at 11.2 kg/ha. Plots were furrow irrigated. To maximize boll opening, cotton fiber quality, and yields, plots were treated with defoliant (thidiazuron and diuron; Ginstar EC, Bayer CropScience, Research Triangle Park, NC, USA) and boll opener (ethephon; Boll Buster, Loveland Products, Inc., Greeley, CO, USA). Cotton boll samples were collected by hand on 6 October 2014 and 2 October 2015. Cottonseeds were processed on a standard saw gin, and seeds were acid-delinted prior to micro-nutrient analyses.

### 3.2. Experimental Design and Statistical Analysis

The experiment was a randomized complete block design with three replicates. Analysis of variance was conducted by PROC MIXED in SAS (SAS, SAS Institute, 2002–2010, Cary, NC, USA) [47]. Year and genotype were considered as fixed effects. Rep (year) was considered as a random factor. The residuals refer to Restricted Maximum Residual Likelihood (REML) values [20,21], and reflect the total variance of the random parameters in the model. Means between lines were separated using Fisher’s protected least significant difference test at significant level of 5% using SAS. PROC CORR in SAS was used for correlations. Results were presented by each year, as year by genotype interactions were significant for some seed composition constituents.

### 3.3. Soil Nutrients Analysis

Random samples were collected from across the field in 2014 and 2015. Soil analysis for nutrient levels in soil was conducted by inductively coupled plasma mass spectrometry (ICP-MS) (Thermo Jarrell-Ash Model 61E ICP and Thermo Jarrell-Ash Autosampler 300 (C Jarrell-Ash Corporation, Waltham, MA, USA) as previously detailed [48]. Briefly, a sample of 5 g soil:20 mL Mehlich-1 solution was used for analysis. Analysis of N, S, and C were based on the Pregl–Dumas method [49,50,51] using a C/N/S elemental analyzer with thermal conductivity cells (LECOCNS-2000 elemental analyzer, LECO Corporation, St. Joseph, MI, USA). Oxygen atmosphere at 1350 ˚C was used to combust soil samples and to convert elemental N, S, and C into N_2_, SO_2_, and CO_2_ gases, respectively. The content of N, S, and C in soil was analyzed by the elemental analyzer as previously detailed (Bellaloui et al., 2015b). Soil analysis showed no nutrient deficiencies in soil. The following are averages of nutrient content in soil across the field: C = 1.02%, N = 0.11%; and (g∙kg^−1^) *p* = 0.289, K = 2.13, S = 0.084, Ca = 4.1, Mg = 3.0, and Fe = 21.03; and (mg∙kg^−1^) B = 1.8; Cu = 15.2; and Zn = 62.5. Organic matter in soil was 2.87%. The crop did not show any nutrient deficiency symptoms under these conditions. Crop management recommendations for the Mississippi Delta region for cotton production were used [52].

### 3.4. Seed Micro-Nutrients Analysis

Dried ground cottonseed samples were analyzed for micro-nutrients Mn, Cu, Zn, and Ni. The samples were processed by digesting 0.6 g in HNO_3_ in a microwave digestion system. Samples were ground using a Laboratory Mill 3600 (Perten, Springfield, IL, USA), and the concentration of Mn, Cu, Zn, and Ni was determined using inductively coupled plasma spectrometry (Thermo Jarrell-Ash Model 61E ICP and Thermo Jarrell-Ash Autosampler 300) [11,37,48]. The concentration of B and Fe was determined as described below.

### 3.5. Boron Analysis

The concentration of boron in cottonseeds was determined according to [53] using the azomethine-H method, and samples were prepared according to [54]. Boron concentration was determined spectrophotometrically using a Beckman Coulter DU 800 spectrophotometer (Fullerton, CA, USA) by reading the samples at 420 nm. The concentration of B was measured after color development, and B concentration was expressed as mg B kg^−1^ dwt.

### 3.6. Iron Analysis

The concentration of Fe in mature cottonseeds were determined according to [55,56]. The concentration was determined by acid wet digestion, extraction, and reaction of the reduced ferrous Fe using 10 mL of 0.02 M 1,10-phenanthroline. The samples were prepared for measurement using a quinol solution of 1% (*w*/*v*) reagent. The concentration of Fe in cottonseeds was measured spectrophotometrically using a Beckman Coulter DU 800 spectrophotometer (Fullerton, CA, USA) by reading the samples at 510 nm. The concentration of Fe was expressed as mg Fe kg^−1^ dwt.

## 4. Conclusions

The current research demonstrated that leaf shape, in our case curly leaf, can partially alter cottonseed micro-nutrients, and the curly leaf trait partially resulted in lower cottonseed micro-nutrients B, Cu, Fe, and Ni. The opposite correlations shown in Zn and Mn could be due to environmental differences in each year, including heat, as 2015 was warmer than 2014. Significant variations of micro-nutrients’ content among cotton lines provides breeders the opportunity to select for cottonseed nutritional qualities. Curly leaf trait could be used as a tool to study leaf development, and physiological, biochemical, and morphological markers underlining the trait.

Future research is needed to further investigate the effect of nutrients supply at different rates on nutrients uptake, distribution, assimilation, seed yield, lint yield, and boll yield. This research will be conducted under controlled conditions so that the contribution of the leaf trait (curly leaf) will be clearly assessed.

## Figures and Tables

**Figure 1 plants-10-01701-f001:**
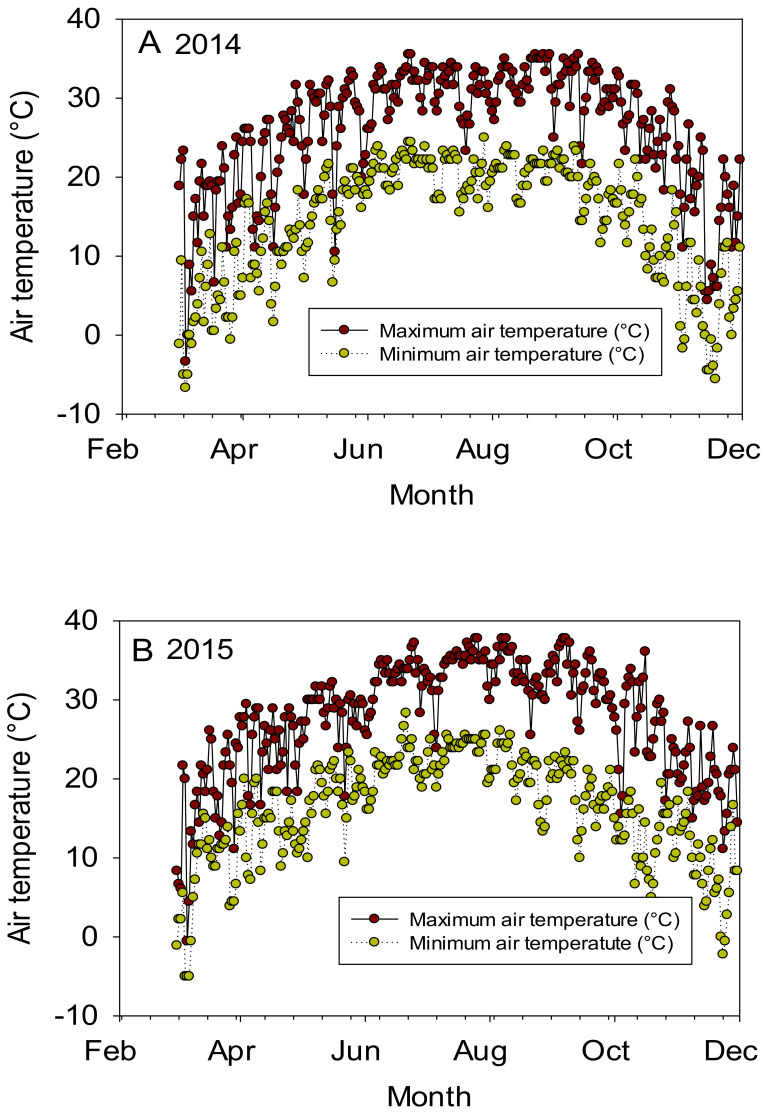
Air temperature (°C) during the growing season in 2014 (**A**) and 2015 (**B**). The experiment was conducted in 2014 and 2015 in Stoneville, MS, USA. Source: Mississippi State University Extension, Delta Weather Center. http://deltaweather.extension.msstate.edu/weather-station-result/ (accessed on 19 August 2021).

**Figure 2 plants-10-01701-f002:**
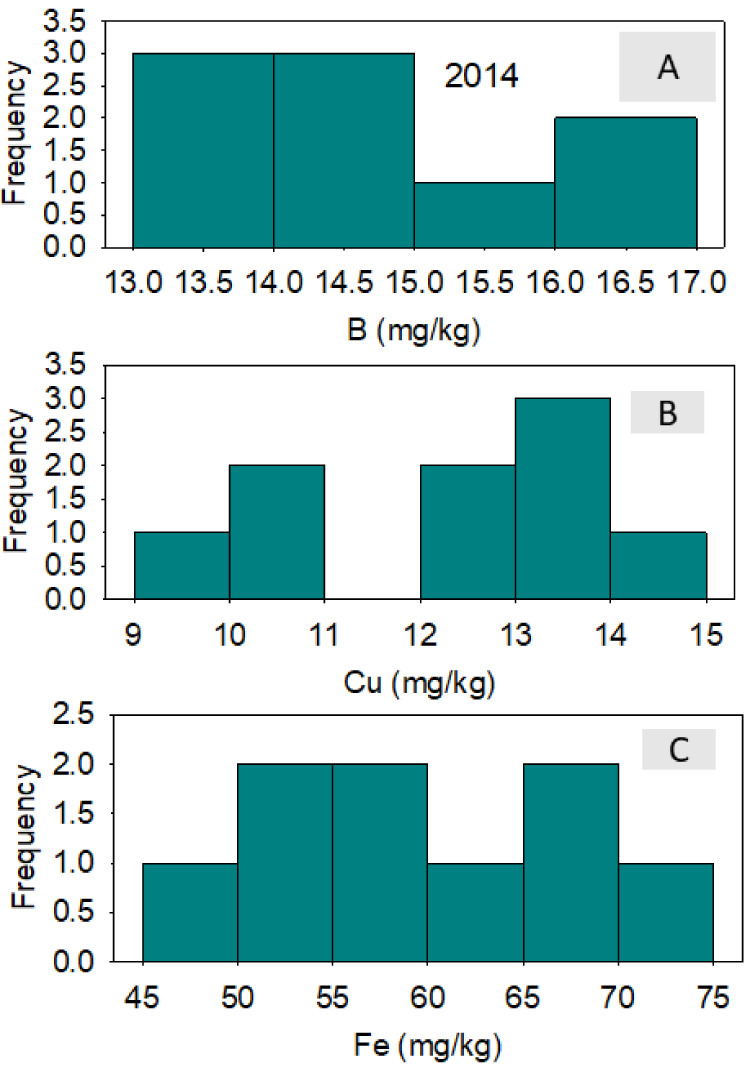
Distributions of cottonseed boron (**A**), copper (**B**), and iron (**C**) across cotton lines. The experiment was conducted in 2014 in Stoneville, MS, USA.

**Figure 3 plants-10-01701-f003:**
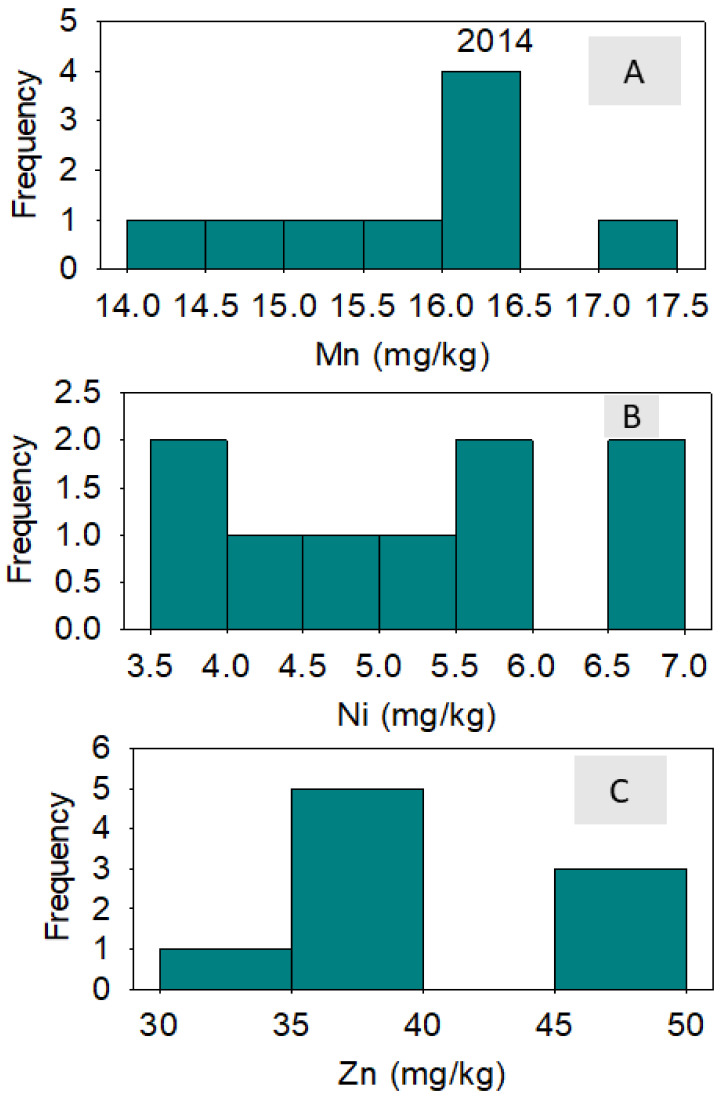
Distributions of cottonseed manganese (**A**), nickel (**B**), and zinc (**C**) across cotton lines. The experiment was conducted in 2014 in Stoneville, MS, USA.

**Figure 4 plants-10-01701-f004:**
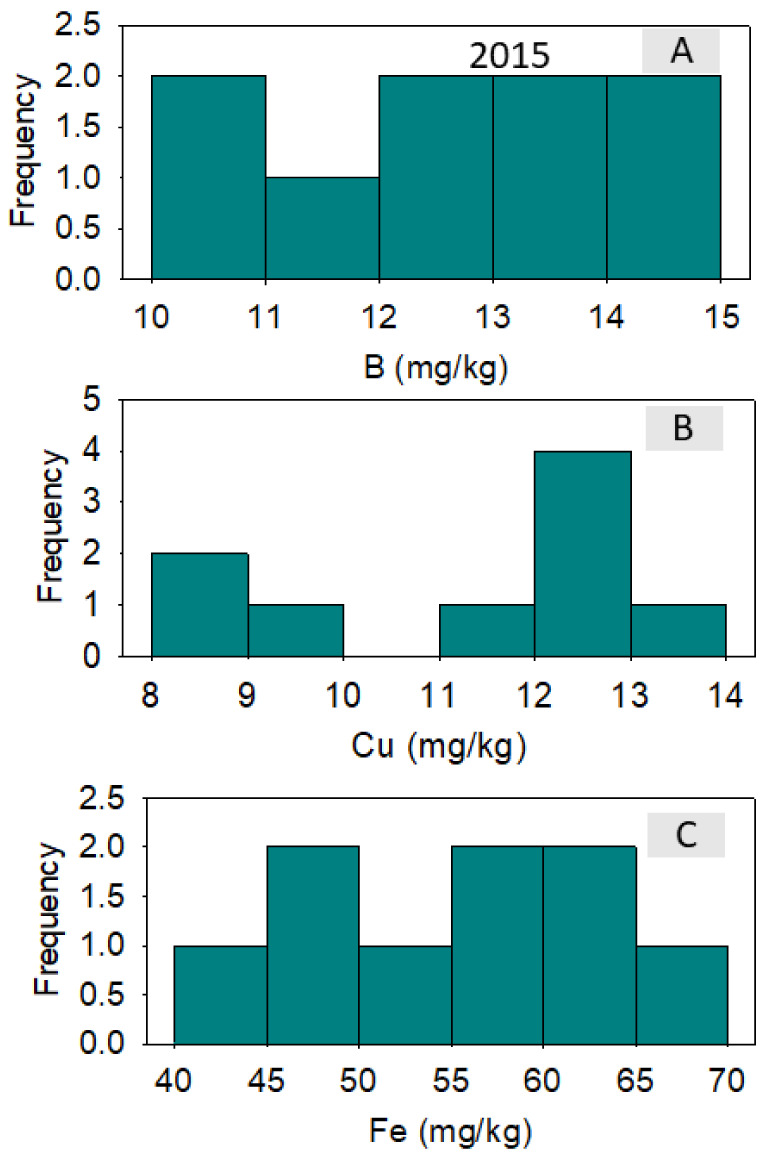
Distributions of cottonseed boron (**A**), copper (**B**), and iron (**C**) across cotton lines. The experiment was conducted in 2015 in Stoneville, MS, USA.

**Figure 5 plants-10-01701-f005:**
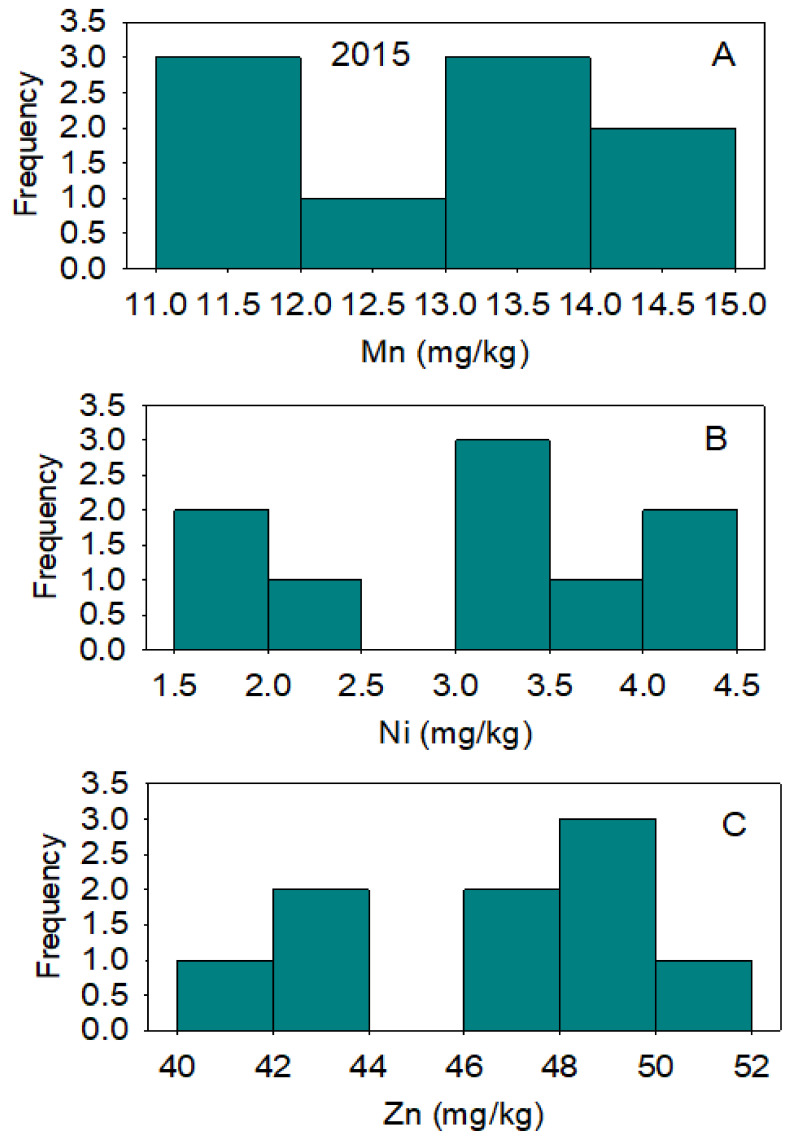
Distributions of cottonseed manganese (**A**), nickel (**B**), and zinc (**C**) across cotton lines. The experiment was conducted in 2015 in Stoneville, MS, USA.

**Figure 6 plants-10-01701-f006:**
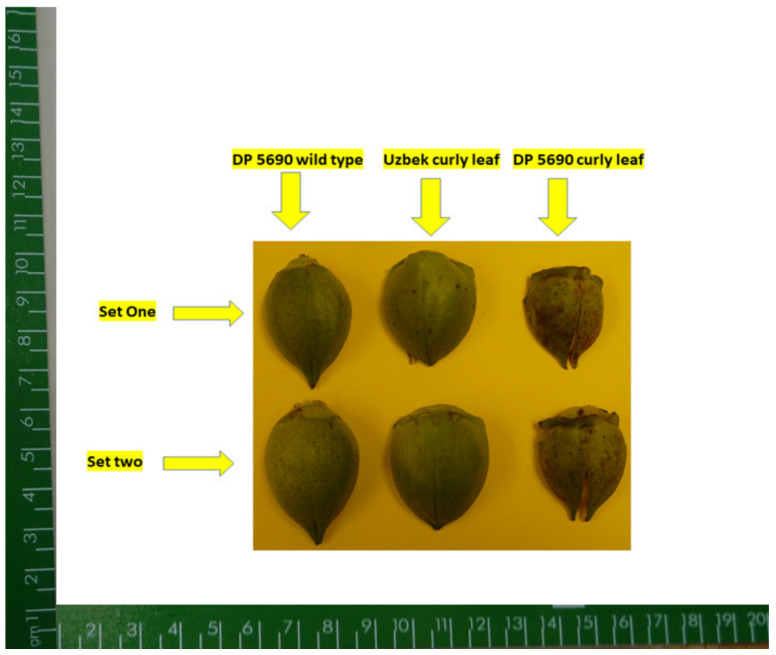
Two sets (replicates) of cotton bolls in DP 5690 wild-type (**left**), Uzbek CRL (**middle**), and DP 5690 CRL (**right**), showing the largest cotton bolls in Uzbek CRL, followed by wild-type, then DP 5690 CRL (smallest). The picture was taken from a height of 30.48 cm.

**Figure 7 plants-10-01701-f007:**
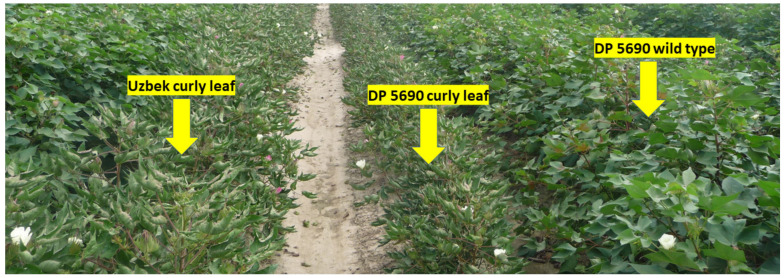
Uzbek curly leaf (higher biomass growth than DP 5690 CRL), DP 5690 wild-type (highest biomass growth), and DP 5690 curly leaf lines (smaller, narrow, rolled-up leaves; less biomass growth) in the field.

**Table 1 plants-10-01701-t001:** Analysis of variance (*F* and *P* values ^a^) for the effect of leaf shape trait (curly leaf) on cottonseed micro-nutrients (mg/kg) boron (B), copper (Cu), iron (Fe), manganese (Mn), nickel (Ni), and zinc (Zn), are essential. The experiment was conducted in 2014 and 2015 in Stoneville, MS, USA.

	DF	B	Cu	Fe	Mn	Ni	Zn
Year	1	77.57 ***	12.9 *	12.64 **	121 ***	53.27 ***	69.27 ***
Line	2	55.38 ***	70.88 ***	75.65 ***	13.49 ***	30.30 ***	2.74 ns
Year *Line	2	1.65 ns	0.75 ns	0.48 ns	22.73 ***	1.06 ns	40.55 ***
Residuals		0.27	0.34	6.79	0.24	0.22	2.82

* Significance at *p* ≤ 0.05; ** significance at *p* ≤ 0.01; *** significance at *p* ≤ 0.001; and ns = not significant.

**Table 2 plants-10-01701-t002:** Effects of leaf shape trait (curly leaf; CRL) on cottonseed micro-nutrients (mg/kg) boron (B), copper (Cu), iron (Fe), manganese (Mn), nickel (Ni), and zinc (Zn). The experiment was conducted in 2014.

Line	B	Cu	Fe (mg/kg)	Mn	Ni	Zn
DP 5690 Wilde-type	16.16	13.08	69.08	15.94	6.28	36.61
Uzbek CRL	14.48	13.74	58.84	14.72	5.04	36.66
DP 5690 CRL	13.42	10.30	49.55	16.41	4.21	46.11
LSD	0.309	0.406	1.515	0.354	0.343	1.167

LSD = Least Significant Difference test, significant at the 5% level. Within each column, the difference between two values is statistically significant if it equals or exceeds the corresponding LSD.

**Table 3 plants-10-01701-t003:** Effects of leaf shape trait (curly leaf; CRL) on cottonseed micro-nutrients (mg/kg) boron (B), copper (Cu), iron (Fe), manganese (Mn), nickel (Ni), and zinc (Zn). The experiment was conducted in 2015.

Line	B	Cu	Fe (mg/kg)	Mn	Ni	Zn
DP 5690 Wilde-type	14.20	12.03	63.07	14.30	3.90	48.70
Uzbek CRL	12.77	12.53	55.70	12.90	3.32	47.80
DP 5690 CRL	10.67	8.47	46.60	11.41	1.80	42.63
LSD	0.296	0.374	1.54	0.227	0.269	0.757

LSD = Least Significant Difference test, significant at the 5% level. Within each column, the difference between two values is statistically significant if it equals or exceeds the corresponding LSD.

**Table 4 plants-10-01701-t004:** Pearson Correlation Coefficients (*p* and R values) between seed nutrients in the near-isogenic lines (Uzbek curly leaf, DP 5690 wild type, and DP 5690 curly leaf) cotton in 2014. The experiment was conducted in Stoneville, MS, USA.

2014					
	B	Cu	Fe	Mn	Ni
**Cu**	*p* = 0.52056				
	*R* = ns				
**Fe**	*p* = 0.88079	0.73212			
	*R* = **	*			
**Mn**	*p* = 0.00082	−0.69623	−0.19359		
	*R* = ns	*	ns		
**Ni**	*p* = 0.79173	0.626	0.94197	−0.08823	
	*R* = **	ns	***	ns	
**Zn**	*p* = −0.7744	−0.83225	−0.74718	0.58827	−0.5666
	*R* = **	**	*	ns	ns

* Significance at *p* ≤ 0.05; ** significance at *p* ≤ 0.01; *** significance at *p* ≤ 0.001; and ns = not significant.

**Table 5 plants-10-01701-t005:** Pearson Correlation Coefficients (*p* and R values) between seed nutrients in the near-isogenic lines (Uzbek curly leaf, DP 5690 wild type, and DP 5690 curly leaf) cotton in 2015. The experiment was conducted in Stoneville, MS, USA.

2015					
	B	Cu	Fe	Mn	Ni
**Cu**	*p* = 0.80413				
	*R* = **				
**Fe**	*p* = 0.89765	0.72654			
	*R* = **	*			
**Mn**	*p* = 0.93164	0.73328	0.92415		
	*R* = ***	*	***		
**Ni**	*p* = 0.8621	0.9018	0.81475	0.82594	
	*R* = **	***	**	**	
**Zn**	*p* = 0.8435	0.87237	0.80441	0.86862	0.86611
	*R* = **	**	**	**	**

* Significance at *p* ≤ 0.05; ** significance at *p* ≤ 0.01; *** significance at *p* ≤ 0.001; and ns = not significant.

## Data Availability

Data is contained within the article.

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
