# Peer review of "Influence of Curly Leaf Trait on Cottonseed Micro-Nutrient Status in Cotton (Gossypium hirsutum L.) Lines"

_plants, 2021, doi:10.3390/plants10081701_

Round 1

Reviewer 1 Report

In the introduction starting on line 81 when you talk about genes, you need to say somewhere that the gene referenced in 30 are phenotypes in Arabidopsis and not cotton.  You only say this in the next sentence talking about FIE  

Tables 2 and 3 need the SD or SE.  I assume that these are not single measurements.  The methods say three reps were grown.

Why did you not use the Fe measurement from the ICP rather this other methodolgy.  Also you need to define what type of ICP was used.  OES, AES or MS.

Figures 2-5 are not informative as you can not tell which line had which level of each micronutrient. Why did you not show the mean for each of the three lines tested with SD or SE in both years in one graph.  Also the letters in the sub figures are different sizes for some reason.

The paragraph starting on line 224 is not correct.  If Uzbek behaved like the WT line then how does the curled leaf trait matter.  It would be something else in linked to the curled leaf trait.  If Uzbek had similar levels of micronutrients as CRL then the trait could be associated with the micronutrient levels.

Your whole part of the intro and discussion on the genes involved is flawed.  How do you know that the genes in Arabidopsis you cite are the same genes in cotton. on chromosome 15.  You would need to sequence these regions in the three lines to make this comment.   Also you do not present any information on the oil content or other phenotypes you mention as part of this work. 

You also can not say anything about uptake or assimilation of the micronutrients as you only present micronutrient data on the seeds.  If you had data on the leaves then you could say something about assimilation or translocation.  But no direct measurements of this was presented.

Also you should never introduce new data in the discussion.  If this data is unpublished then publish the data either here or in another publication first.  

Did you look at the number of seeds per plant to look at the overall levels of the micronutrients in the plant.  Is the difference in levels seen due to seed number or aleurone layer proportion rather than something involved in uptake or assimilation?

How did you define boll size is it volume or weight?  Also figure 6 needs some type of scale for reference.

Reviewer 2 Report

The present paper "Influence of curly leaf trait on cottonseed micro-nutrient status in cotton (Gossypium hirsutum L.) lines" addresses a subject of interest and is well written and designed.

The objective of this research was to investigate the influence of leaf shape on cotton seed micro- nutrients content and the aim of the study is achieving with the presentation and discussion of results.

I would like to know how are you secured that the acid delinting of seeds did not affect the micronutrients content of the seeds?

 Also the following aspects have to be clarified.

In Figure 2 and 4  titles after the word iron in the brackets has to be (C ) and not (Fe)

The line 295-297 was the same with 310-311.

In Figure 7 the B in the figure is not necessary.

If the above aspects are better presented and supported, the paper is sufficiently novel and contains very interesting data for publication.

Author Response

Reviewer comment

The present paper "Influence of curly leaf trait on cottonseed micro-nutrient status in cotton (Gossypium hirsutum L.) lines" addresses a subject of interest and is well written and designed.

The objective of this research was to investigate the influence of leaf shape on cotton seed micro- nutrients content and the aim of the study is achieving with the presentation and discussion of results.

I would like to know how are you secured that the acid delinting of seeds did not affect the micronutrients content of the seeds?

Authors response

Very good question. We cannot be secured 100% that the delinting does not have an effect. However, this procedure is standard and acceptable by all cotton scientists. Even though, if there was an effect, it should affect all the seeds of all lines; therefore, differences between nutrients in lines should be reflected anyway. I think it is a good idea in the future to check the effect of de-linting types on macro- and micro-nutrients. We appreciate the comment to use it in the future research.

Reviewer comment

Also the following aspects have to be clarified.

In Figure 2 and 4 titles after the word iron in the brackets has to be (C ) and not (Fe)

Authors response

Revised and corrected as suggested.

Reviewer comment

The line 295-297 was the same with 310-311.

Authors response

Revised and corrected. The formatting issue could be solved by the editorial technical team, and if they have issues with it, we will be happy to do our best to fix it.

Reviewer comment

In Figure 7 the B in the figure is not necessary.

Authors response

Revised and fixed

Reviewer comment

If the above aspects are better presented and supported, the paper is sufficiently novel and contains very interesting data for publication.

Authors response

We appreciate the review and comments.

Reviewer 3 Report

My opinion is that this article is suitable for publication in Plants as it stands.

The article investigates an important topic ie how mineral contents of cotton seeds are modulated by genotype, and in particular by leaf morphology. Further, interesting data are presented regarding the effect of growth conditions.

The experiments are well planned and executed.

The paper is well presented - the English is good and the flow of the paper allows one to read it easily.

The paper will be of interest to a range of researchers

Author Response

Reviewer comment

My opinion is that this article is suitable for publication in Plants as it stands.

The article investigates an important topic ie how mineral contents of cotton seeds are modulated by genotype, and in particular by leaf morphology. Further, interesting data are presented regarding the effect of growth conditions.

The experiments are well planned and executed.

The paper is well presented - the English is good and the flow of the paper allows one to read it easily.

The paper will be of interest to a range of researchers

Authors response

We appreciate the positive comments of the reviewer

Reviewer 4 Report

Comments and Suggestions for Authors

Influence of curly leaf trait on cottonseed micro-nutrient status in cotton (Gossypium hirsutum L.) lines

This manuscript is very well written and organized. The subject is interesting and fall within the scopoe of the journal. The experimental dataset undoubtedly are useful  and constitutes some scientific values. The presented manuscript deals with the current problem of optimal micronutrient content. Manuscript presented the results research the influence of leaf shape on cottonseed micro-nutrients content using two isogenic cotton lines differing.

The metodology of the article is very good, the methods were chosen well. The results are clearly presented and supported by arguments.

The following points may be addressed by the Authors to enhance the usefulness of the article.

Remarks

  1. Macronutrients have not been studied. Lines 107-108 – delete the sentence.
  2. Description of figure 1-5 adjust the font and size to the publishing requirements.
  3. Table 1: 77.57***a a - ? (LSD?). Explain below the table.
  4. In the Results and discussion section, I propose to include a subsection on the occurrence of the phenomenon of ionic antagonism and synergism between the analyzed micronutrients.
  5. Subsection 3.3. - The soil pH should be supplemented. In my opinion, pH is important for the absorption of micronutrients.

The entire manuscript must be adapted to the publishing requirements.

I recommend this manuscript for publication.

Author Response

Reviewer comment

This manuscript is very well written and organized. The subject is interesting and fall within the scope of the journal. The experimental dataset undoubtedly are useful and constitutes some scientific values. The presented manuscript deals with the current problem of optimal micronutrient content. Manuscript presented the results research the influence of leaf shape on cottonseed micro-nutrients content using two isogenic cotton lines differing.

The methodology of the article is very good, the methods were chosen well. The results are clearly presented and supported by arguments.

The following points may be addressed by the Authors to enhance the usefulness of the article.

Authors response

We appreciate the positive comment of the reviewer.

Remarks

  1. Macronutrients have not been studied. Lines 107-108 – delete the sentence.

Deleted

  1. Description of figure 1-5 adjust the font and size to the publishing requirements.

Authors response: It should be acceptable for higher resolution; we have been producing these type of graphs to make sure they have a good quality once published. If the publishing service requires that, we will be happy to do whatever they ask us to do.

  1. Reviewer comment: Table 1: 77.57***a a - ? (LSD?). Explain below the table.

Authors response: It looks like it was confusing. Therefore, we deleted the superscript “a” and wrote a normal footnote to explain the level of significance “*, **, and ***” and LSD.

  1. Reviewer comment: In the Results and discussion section, I propose to include a subsection on the occurrence of the phenomenon of ionic antagonism and synergism between the analyzed micronutrients.

Authors response: Thank you for the comment. We included a paragraph in the discussion section where we discussion the relationships between micronutrients. This would be beneficial to the reader where it occurred. We appreciate the suggestion.

  1. Reviewer comment: Subsection 3.3. - The soil pH should be supplemented. In my opinion, pH is important for the absorption of micronutrients.

Authors response: Our soil pH is neutral so no toxicity or deficiency occurred on the leaves or the crop in general.

Reviewer comment: The entire manuscript must be adapted to the publishing requirements.

Authors response: We revised the manuscript to make sure it fits the requirements

Reviewer comment: I recommend this manuscript for publication.

Authors response: We appreciate the reviewer’s positive comment.

Reviewer 5 Report

1. The abstract can serve as a stand-alone document that succinctly describes both procedures and conclusions. 
2. The introduction is informative, precise, and comprised of relevant content. The literary structure of the introduction is also good. 
3. The experimental methodology was simple but appropriate and scientific, and the analyses were done correctly.  
Authors expected that leaf shape can alter cottonseed micronutrients status. They tried to correlate the leaf shape traits to study leaf development, physiological, biochemical, and morphological processes, and also microelements content. They used two near-isogenic cotton lines differing in curly leaf in two years field experiment.
Despite the idea behind the study of the role of the curling leaves to the mineral composition of the seeds in cotton, they are evident many mechanisms can influence accumulation processes from the uptake of mineral elements to accumulation activities of the sink cells, is important to compare this idea also on the different crop species.
4. The presentation of the results in the form of graphs and tables is clear and the results are well described. All results are comprehensible. The discussion is sufficient and the conclusions are well formulated and justified. The figures, captations content are considerable, readable, and useful. The understanding of mechanisms is limited, as it is restricted to papers that have a particular view and deliberately ignore alternatives, and does not present a balanced view of the evidence. But paper presents novel results.
Authors demonstrated that leaf shape, curly leaf, can partially alter cottonseed micro-nutrients, and curly-leaf trait partially resulted in lower cottonseed micro-nutrients B, Cu, Fe, and Ni.
5. The discussion lacks a bit in-depth. Discussion and the writing should be improved to make the manuscript easy to follow. The paper will be more attractive.
6. Avoid writing long paragraphs.
7.  The manuscript has to explain the novelty of this work better, mostly plant regulation mechanisms in mineral nutrition and also the role of the high temperature and other environmental stress on both genotypes on the leaves shape. What are the transpiration and relationships between transpiration and mineral nutrition processes?
Authors could add new conclusions, future perspectives. the body of the text has to be improved in the discussion. I would have expected a more critical discussion of the results. Some arguments need a simpler, clearer, and tighter presentation.
8.  I suggest adding also new references to support the interpretations.
9. Authors have to unify the style of references according to the journal rules. 

Author Response

  1. The abstract can serve as a stand-alone document that succinctly describes both procedures and conclusions.
  2. The introduction is informative, precise, and comprised of relevant content. The literary structure of the introduction is also good.
  3. The experimental methodology was simple but appropriate and scientific, and the analyses were done correctly.

Authors expected that leaf shape can alter cottonseed micronutrients status. They tried to correlate the leaf shape traits to study leaf development, physiological, biochemical, and morphological processes, and also microelements content. They used two near-isogenic cotton lines differing in curly leaf in two years field experiment.

Despite the idea behind the study of the role of the curling leaves to the mineral composition of the seeds in cotton, they are evident many mechanisms can influence accumulation processes from the uptake of mineral elements to accumulation activities of the sink cells, is important to compare this idea also on the different crop species.

Authors response: Great suggestion, and we appreciate it

  1. The presentation of the results in the form of graphs and tables is clear and the results are well described. All results are comprehensible. The discussion is sufficient and the conclusions are well formulated and justified. The figures, captations content are considerable, readable, and useful. The understanding of mechanisms is limited, as it is restricted to papers that have a particular view and deliberately ignore alternatives, and does not present a balanced view of the evidence. But paper presents novel results.

Authors response: Very good point. We are hoping to study the photosynthesis process with mechanistic approach in our future paper on some metabolites. We also, hope to study the uptake, distribution, and accumulation in these lines using the nutrient supply approach under sufficient and deficient nutrients supply.

Authors demonstrated that leaf shape, curly leaf, can partially alter cottonseed micro-nutrients, and curly-leaf trait partially resulted in lower cottonseed micro-nutrients B, Cu, Fe, and Ni.

  1. The discussion lacks a bit in-depth. Discussion and the writing should be improved to make the manuscript easy to follow. The paper will be more attractive.

Authors response: We revised the manuscript as suggested, taking in account the reviewer’s suggestion. We would like to expand to uptake, assimilation, and distribution (mechanistic approach), but some other reviewers may not like it as we did not have real data to support our claim. We appreciate the reviewer’s point to consider it in our future research.

  1. Avoid writing long paragraphs.

Authors response: The manuscript was revised in case there are some unclear sentences.

  1. The manuscript has to explain the novelty of this work better, mostly plant regulation mechanisms in mineral nutrition and also the role of the high temperature and other environmental stress on both genotypes on the leaves shape. What are the transpiration and relationships between transpiration and mineral nutrition processes?

Authors response: As the reviewer indicated, environmental factors such rainfall, temperature, and the rate of transpiration and their interactions with mineral nutrition process are important; however, without real experiment and real data relating these factors to nutrient uptake and assimilation may not be attracted by the reader.

Authors could add new conclusions, future perspectives. the body of the text has to be improved in the discussion. I would have expected a more critical discussion of the results. Some arguments need a simpler, clearer, and tighter presentation.

Authors response: Great point about future research on these lines. We added a statement in the conclusion, and we revised the entire manuscript to respond to the reviewer’s comment.

  1. I suggest adding also new references to support the interpretations.

Authors response: We believe the references used is adequate otherwise it will make the manuscript bigger and more complex.

  1. Authors have to unify the style of references according to the journal rules.

Authors response: We revised the references and make sure they are in compliance with the journal instructions.

Round 2

Reviewer 1 Report

The authors have basically decided not to alter change most of the comments I have previously made.  Some like the response to measurements of Fe make no sense as ICP-MS will measure 40 or so elements all at once.  So there was no reason to run another Fe measurement. 

As for the figures at least color each genotype to give the reader some understanding of which genotypes had high or low levels of each nutrient.  The response was it was used for breeding purposes, but you need to know which lines to cross to improve a given trait.  Also why is figure 4 now have letters with gray boxes but none of the other figures?

The new discussion talking about the roles of the various elements on physiological traits need citations.  Starting on line 293.

Your blurb in the Fe methods section line 493 is not a method.  Also this statement is not true.  As you show in the picture of the three genotypes plant height, and seed pods are also different between the lines so you can not say that curly leaves are the only difference. 

For figure 6 it is not hard to use the same set up and take a picture of a ruler then add a black bar somewhere in the figure to give some sense of scale.  If a person does not know the size of a cotton seed pod they have no idea about the difference.

Reviewer 5 Report

I checked the status, reviews, authors' comments on this paper.

The authors have made a great effort in introducing amendments to the manuscript in response to all the reviews. I am satisfied with the revised version, which in my opinion is ready for publication.

Author Response

We thank the reviewer for the positive comment, and appreciate the time and effort.

Round 3

Reviewer 1 Report

The frequency figures do not really add anything to the m/s